# Addressing the Gaps in Post-Stroke Sexual Activity Rehabilitation: Patient Perspectives

**DOI:** 10.3390/healthcare7010025

**Published:** 2019-02-05

**Authors:** Sarah Prior, Nicole Reeves, Gregory Peterson, Linda Jaffray, Steven Campbell

**Affiliations:** 1School of Medicine, College of Health and Medicine, University of Tasmania, Hobart, Tasmania 7000, Australia; Nicole.reeves@utas.edu.au (N.R.); g.peterson@utas.edu.au (G.P.); linda.jaffray@utas.edu.au (L.J.); 2School of Health Sciences, College of Health and Medicine, University of Tasmania, Newnham Drive, Newnham, Tasmania 7250, Australia; Steven.campbell@utas.edu.au

**Keywords:** stroke, rehabilitation, sexuality, sexual dysfunction, education, personal satisfaction

## Abstract

Sexual dysfunction is common but often under-recognised or neglected after stroke. This study sought to identify the existing methods for providing information and discussion on post-stroke sexual activity, and perceived gaps from the patient perspective. A sample of 1265 participants who had been admitted to any of the four major public hospitals in Tasmania, Australia, with stroke (International Classification of Diseases (ICD-10) group B70) were mailed a survey assessing their experiences with, and opinions about, receipt of post-stroke sexual activity education. One hundred and eighty-three participants (14.5%) responded; of these, 65% were male and the mean age was 69.1 years. The results indicated that, whilst over 30% or participants wanted to receive information related to post-stroke sexual activity, only a small proportion of participants (8.2%) had received this. In terms of the method of receiving this information, participants preferred to receive this from a doctor in a private discussion with or without their partner present. The delivery of post-stroke sexual activity information and education is inconsistent and fails to meet patient needs within major Tasmanian hospitals, highlighting the importance of developing sound, routine, post-stroke education and information processes.

## 1. Introduction

Sexual dysfunction after stroke is common. Problems with sexual activity in post-stroke patients are complex and often due to multiple aetiologies, including both physical and psychosocial factors [1,2,3,4]. One Turkish study reported that two-thirds of the participants did not resume sexual activity by 6 months following a stroke and that sexual satisfaction decreased in the remaining participants; three-quarters of the participants regarded sexual functioning as an important factor in rehabilitation [5]. 

Counselling of, or education for, stroke patients in rehabilitation around sexual activity can be a challenging experience, particularly for unprepared staff, but it is a necessity for improving and/or maintaining quality of life. Giaquinto et al. [6] studied quality of life after stroke, and suggested that sexual activity should be carefully evaluated in post-stroke rehabilitation and that intervention is often necessary to offset psychological disturbances, lack of information and prejudice. As interpersonal problems are recognised, couples’ sexual activity counselling is important for a return to a quality of life similar to that pre-stroke. 

According to the Australian Stroke Foundation Clinical Guidelines [7], stroke rehabilitation should address the issue of post-stroke sexual activity prior to patient discharge from hospital. The opportunity should be provided for stroke survivors and their partners to discuss issues related to sexual activity with an appropriate health professional, along with the provision of written information to take home. Furthermore, any interventions should take care to address psychosocial aspects, in addition to issues of physical function. 

Several studies have examined satisfaction with patient education and counselling related to post-stroke sexual activity and also the reasons for sexual dissatisfaction and dysfunction following a stroke [8,9]. Some of the main issues identified are as follows: (1) sexual activity is an integral part of the rehabilitation process; (2) sexual activity/regaining full sexual function is not considered a routine rehabilitation goal; (3) sexual activity issues are not adequately addressed by healthcare staff; (4) there are no clear guidelines as to who is responsible for addressing sexual activity with patients; (5) healthcare staff are not provided adequate training in addressing sexual activity issues; (6) sexual health needs are affected by cultural factors; and (7) the most common factor for not resuming sexual activity after stroke is fear of elevating blood pressure and precipitating another stroke. 

The aims of this study were to identify the existing methods for providing information and discussion on post-stroke sexual activity, and perceived gaps from the patient perspective. We suspected that current protocols for delivering information and education around post-stroke sexual activity may be inconsistent across the four major public hospitals in Tasmania, Australia, and may not meet patient needs.

## 2. Methods

### 2.1. Participants

One thousand two hundred and sixty-five patients who were admitted to a major Tasmanian public hospital with an International Classification of Diseases (ICD-10) group B70 diagnosis (Stroke and other cerebrovascular disorder) between 1 January 2014 and 31 December 2016 were eligible for participation in this study. The patients were identified from electronic hospital records. The four hospitals involved in this study were Royal Hobart Hospital, Launceston General Hospital, Mersey Community Hospital (Latrobe), and North West Regional Hospital (Burnie). 

### 2.2. Materials

A 14-item self-report survey was used to ask patients about their experience with, and opinions of, post-stroke sexual activity education, and the type of information that they believe would be helpful and appropriate for future stroke patients. The questions were based on current Australian stroke guideline recommendations and previous research with post-stroke patients. 

### 2.3. Procedure

All eligible patients were mailed an information sheet and survey with a postage-paid, return addressed envelope. They were requested to read the information sheet, complete the anonymous survey and return within 8 weeks. Data analysis was descriptive (e.g., calculation of percentages) and performed using Microsoft Office Excel (2016). Ethical approval was obtained from the Tasmanian Health and Human Research Ethics Committee (H0016296).

## 3. Results

One hundred and eighty-three patients (14.5%) returned completed surveys; of these, 65% were male and the mean (SD) age for all participants was 69.1 (12.8) years (range: 34–98 years).

### 3.1. Patient Experience 

Only 15 participants (8.2%) indicated that they had received information relating to post-stroke sexual activity, with 12 of these (85%) receiving this information whilst still in hospital. Of the latter, eight participants received this information in written form, three verbally and one a combination of both methods. This information was delivered by a doctor (1), a nurse (9), both a doctor and nurse (1) and other channels, including volunteer stroke foundation (1), and the stroke foundation website (1). The remaining three participants received this information following discharge, with one participant finding the information online, one obtaining the information from their doctor and the other utilising both of these information sources. One participant indicated that they did not receive any information regarding post-stroke sexual activity but also answered that they looked up their own information online post discharge.

Of the 15 participants who received information post-stroke, only one-third believed that it met their needs (Table 1). When asked whether the participant’s spouse had received any information related to post-stroke sexual activity, 153 (84%) answered no, two answered yes, and the remaining 28 (15%) did not answer.

### 3.2. Patient Opinions

The results indicated that 63 participants (34%) wanted to receive information about sexual activity after stroke. Fifty-seven (31%) participants indicated that they did not want to receive any information about sexual activity after stroke, 45 (25%) were not bothered, and 18 did not answer this question. 

One hundred and twenty-five (68%) participants felt that all stroke patients should be routinely offered information about sexual activity after stroke. Nine participants did not think this was necessary, 31 (17%) were not bothered, and 20 (11%) did not answer this question. One hundred and twenty-seven (69%) participants felt that spouses/partners should also be routinely offered information about post-stroke sexual activity. Eight participants did not believe that spouses/partners should be offered this information, 28 (15%) were not bothered, and 21 (11%) did not answer this question. 

Fifty-four (30%) participants preferred information in a written format, 31 (17%) preferred to have someone speak to them verbally and 69 (38%) would prefer both methods of communication. Six participants answered ‘other’ and indicated that they would prefer information via video (1), online (2), none (1), and not bothered (2). Twenty-five (14%) participants did not answer this question. 

As indicated in Table 2, 40 (22%) participants stated that they would prefer to receive information about sexual activity post-stroke from a doctor and 37 (20%) would like all clinical staff to be able to provide this information. When asked which method of information delivery is best, 30 (16%) participants wanted their partners to be involved in the discussions, 22 (12%) preferred a brochure or pamphlet, and 20 (11%) wanted private discussions (Table 3).

A number of issues were considered to be important to include in information about sexual activity post-stroke (Table 4). A section allowing for participants to provide their own comments regarding the survey or their own experience post-stroke suggested a number of improvements. Some of those comments included:
“I wasn’t worried about receiving no information about sexual activity as I was not sexually active and did not have a partner. I’m sure it would be very different for other people—especially people who don’t feel confident asking questions.”
“I think this subject is very important because I lost sexual drive after stroke which split my family up my wife went somewhere else for sex so she said that doesn’t need me anymore and kicked me out.”
“Lots of pamphlets in hospitals and doctors but I found none on this topic.”
“My wife asked the doctor before I was discharged for advice as to whether we could continue with intercourse and we were told to speak to our GP; they were not interested in giving us advice in the hospital. By the time we saw the doctor we realised all was ok and continued with our life. But we wish we have had been reassured in the hospital. It is very important that this is discussed.”
“No one even gave a thought about sex; they were more interested in getting me back on my feet and moving again.”
“I wanted to know if it was my medications affecting my libido.”

## 4. Discussion

Post-stroke sexual activity information and education delivery did not meet patient and partner needs in the four major Tasmanian hospitals. We did note that 31% of participants in our study indicated that they did not want to receive any information about post-stroke sexual activity. However, very few patients or their spouses received any information regarding post-stroke sexual activity, and only two-thirds of those who did found it helpful. Participants were generally underwhelmed with the apparent motivation of staff to ensure they were well educated about sexual activity following their stroke. Our findings are consistent with those of a smaller North American study [9], which reported that most stroke rehabilitation programs do not include a consistent approach for addressing sexual activity post-stroke. They found that sexual activity was rated by patients as a moderately to very important issue in context of their overall stroke recovery and rehabilitation. In both studies the primary concern for stroke survivors was maximising the quality of life, with participants feeling that intimate relationships and sexual activity counselling is an important part of their rehabilitation. As noted by Kautz and Van Horn [10], intimacy and sexual concerns are often ignored by the post-stroke rehabilitation team, yet research shows that couples want information to assist them to maintain their sexual relationships. 

Overall, the majority of participants in our study indicated that they felt all of the statements in Table 4, made in relation to post-stroke sexual activity, are important, particularly around the effects of medications. Our results suggest that patients would prefer to receive this type of information from at least one clinical staff member, with doctors being the preferred clinician, although the results also indicated that it is important for all clinical staff to be able to provide this type of information. The participants favoured private discussions with or without their partners present. However, a relatively large percentage of participants still preferred to have a brochure or pamphlet. This may reflect the sensitive nature of the topic of sexual activity from a patient’s perspective and discomfort in asking for this information. 

The clinical responsibility of discussions around post-stroke sexual activity are ambiguous and the development of clearer processes would be beneficial to patients, their spouses and for staff involved in stroke rehabilitation. McLaughlin and Cregan [11] utilised questionnaires to gather information of the experiences of staff with patients seeking sexual activity advice, their comfort levels in responding to these requests, and their attitudes towards this in the future. The results suggested that the staff had received no training in addressing sexual activity issues with patients following a stroke, as well as very little experience in addressing these issues with patients. Staff indicated that if they were approached by a patient with a sexual activity issue they would refer them onto to a relevant agency with more expertise in this particular area. Staff also indicated that there were a number of reasons why they felt uncomfortable discussing sexual activity with their patients. These reasons included lack of experience, own personal beliefs, fear of offending the patient, embarrassment, perception that it was other professionals’ responsibility, and lack of training. Healthcare staff working with post-stroke patients (patients with disabilities) have previously reported feeling they lacked the knowledge, comfort and confidence to respond to patients’ needs in a sensitive and informed manner [8].

There appears to be considerable variation in the approach to discussing sexual activity after stroke, ranging from a routine discussion early in the course of care to only addressing these issues when the patient or family member specifically raises a concern. Overall findings show that while health professionals are regularly required to address sexual health issues with their patients in their daily practice, a lack of appropriate education and training in this area affects their ability to proactively respond in a comfortable and skilled manner. This suggests that health professionals could benefit from routine education and training around sexual activity post-stroke and sexual health issues in rehabilitation. 

A Korean study by Song et al. [12] demonstrated that a sexual rehabilitation program for stroke patients and their spouses, with educational content, emotional counselling and specific strategies to minimise or overcome post-stroke sexual activity problems, improved sexual satisfaction and frequency of sexual activity. In a study at a Canadian rehabilitation centre, Guo et al. conducted a gap analysis through staff member interviews and retrospective chart reviews [13]. At baseline, very few post-stroke patients were given the opportunity to discuss sexual health concerns. Changes subsequently implemented included a reminder system, standardisation of care processes for sexual health, patient-centred time points for the delivery of sexual health discussions, and the development of a sexual health supported conversation tool for patients with aphasia. By the end of the 10-month project, the proportion of patients provided with the opportunity to discuss sexual health during inpatient rehabilitation had increased to 80%.

A limitation of the current study is the low response rate, perhaps due to the sensitive nature of the survey, from one State of Australia. Therefore, caution is required in generalising the findings to other populations of stroke patients. 

## 5. Conclusions

There is inadequate and inconsistent delivery of education and information regarding post-stroke sexual activity, and patients have indicated that they were not always comfortable requesting information. Education and training for healthcare staff around sexual activity education and discussion during stroke rehabilitation would be beneficial to help staff engage with patients and their partners on this important topic. Patients and their partners should be encouraged to ask questions around sexual activity and should also have the opportunity to discuss problems post-discharge.

## Figures and Tables

**Table 1 healthcare-07-00025-t001:** Post-stroke information and patient needs.

Information Met All of My Needs	Information Met Most of My Needs	Information Met Some of My Needs	Information Met None of My Needs	Unsure
6 (40%)	1 (7%)	2 (13%)	4 (27%)	2 (13%)

**Table 2 healthcare-07-00025-t002:** Preferred sources of post-stroke sexual activity information.

Source of Post-stroke Sexual Activity Information	Number of Participants (Female/Male)
All clinical sources	37 (14/23)
Allied Health Professional	28 (9/19)
Doctor and Allied Health Professional	25 (5/20)
Doctor	40 (11/29)
Nurse	3 (1/2)
Nurse and Allied Health Professional	5 (4/1)
Nurse and Doctor	9 (4/5)
Found information on their own (e.g., Internet search)	1 (0/1)
Other—Someone with experience	1 (0/1)
Other—Not bothered	1 (0/1)
Other—No answer	1 (1/0)

**Table 3 healthcare-07-00025-t003:** Preferred information delivery method.

Group	Information Delivery	Number of Responses (Female/Male)
A	Private one-on-one discussion	20 (3/17)
B	Discussion with couples (if applicable)	30 (11/19)
C	Group discussions	1 (1/0)
D	Being given a brochure/pamphlet with no discussion	22 (5/17)
	A and B	28 (8/20)
	A and C	2 (1/1)
	A and D	11 (7/4)
	B and C	0 (0/0)
	B and D	9 (2/7)
	C and D	2 (0/2)
	A, B, and C	4 (2/2)
	A, B, and D	11 (5/6)
	B, C, and D	0 (0/0)
	A, C, and D	0 (0/0)
	All methods	10 (4/6)
	Other	1 (0/1)
	Did not answer	32 (10/16) *

* Six participants did not include their gender.

**Table 4 healthcare-07-00025-t004:** Types of information that would be helpful post-stroke.

Statement	Important (F/M)	Not Sure (F/M)	Not Important (F/M)	No Answer (F/M)
Dealing with physical limitations and changes to sexual activity after stroke	122 (40/81) *	15 (4/11)	13 (6/7)	33 (10/22) *
How medications might affect sexual activity after stroke	128 (41/85) *	13 (5/8)	8 (4/4)	34 (9/25)
Dealing with the effect stroke had on your feelings and thoughts about your sex life	119 (40/77) *	23 (4/19)	10 (5/5)	31 (11/20)
Knowing who you can talk to about sexual issues after stroke	119 (38/78)	17 (5/12)	16 (6/9) *	31 (10/20) *
How interest in sex might change after stroke, for either yourself or your partner	118 (38/80)	23 (7/16)	12 (5/7)	30 (9/19) *
Information about how to talk to your partner about sexual issues after stroke	109 (39/70)	25 (6/19)	17 (3/14)	32 (10/20) *
Dealing with worry or concerns about sex causing another stroke	100 (34/65) *	31 (7/24)	21 (8/13)	31 (10/20) *

* Gender was not supplied for all responses. F/M: female/male.

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
