# Peer review of "Addressing the Gaps in Post-Stroke Sexual Activity Rehabilitation: Patient Perspectives"

_healthcare, 2019, doi:10.3390/healthcare7010025_

Round 1

Reviewer 1 Report

Prior et al presented an interesting topic which may attract general attentions. I only have a few minor questions:

(1) Bar chart figures are highly recommended to present the results instead of tables.

(2) It would be more rigor to separate the male from female and present the data with several age groups (34-45 y/o, 45-60y/o and 60-98 y/o).

(3) Bigger sample size would be more convincing.

Author Response

Reviewer 1 Comments:

Prior et al presented an interesting topic which may attract general attentions. I only have a few minor questions:

(1) Bar chart figures are highly recommended to present the results instead of tables.

(2) It would be more rigor to separate the male from female and present the data with several age groups (34-45 y/o, 45-60y/o and 60-98 y/o).

(3) Bigger sample size would be more convincing.

Dear Reviewer,

Thank you for your review and your feedback – the authors appreciate the comments. To address your concerns please find our responses below.

1.       Bar charts are certainly a great way to present data, however, in this manuscript we feel that being consistent is important to our readers and some of the information (eg. Table 3 and Table 4) is too “busy” to present in a bar graph.

2.       Given the small sample numbers for Table 1, adding in gender and age group would add little value. In Tables 2,3 and 4, we have included gender differences to provide more rigour but not age groups as the majority of our participants were in the 60-98 bracket.

3.       We agree that a larger sample size would have been more beneficial to the study. As the population in Tasmania is relatively small and the post-stroke population generally elderly, the response rate was smaller than expected, which was mentioned in our discussion (Line 205). A nationwide survey would be great as a future study.

Reviewer 2 Report

The authors address an important issue of sexual activity post-stroke. Even though sexual dysfunction is common, it is an issue not sufficiently addressed as part of post-stroke care and rehabilitation. The study focused on perceived gaps in sexual activity information and education that is provided post-stroke from a patients perspective and concludes that the delivery of such information is inadequate and not consistent. The study is timely considering it is an important issue that has gone unregulated and not addressed with necessary significance. The manuscript is well written and includes important concerns in the questionnaire as suggested in the Australian Stroke Foundation Clinical Guidelines.

Minor concern:

The authors report that 183 participants responded and that 65% were males. The authors did not provide specific age range for females. It would be interesting to know if there is any significant difference in the perception of sexual activity education between male and female post-stroke participants.

Author Response

The authors address an important issue of sexual activity post-stroke. Even though sexual dysfunction is common, it is an issue not sufficiently addressed as part of post-stroke care and rehabilitation. The study focused on perceived gaps in sexual activity information and education that is provided post-stroke from a patients perspective and concludes that the delivery of such information is inadequate and not consistent. The study is timely considering it is an important issue that has gone unregulated and not addressed with necessary significance. The manuscript is well written and includes important concerns in the questionnaire as suggested in the Australian Stroke Foundation Clinical Guidelines.

Minor concern:

The authors report that 183 participants responded and that 65% were males. The authors did not provide specific age range for females. It would be interesting to know if there is any significant difference in the perception of sexual activity education between male and female post-stroke participants.

Dear Reviewer,

Thank you for your review and your feedback. The authors greatly appreciate your time and your comments.

To address your concerns, please find our response below.

1.       We have included the genders of the participant responses in Tables 2,3 and 4 to provide further rigor and more interesting results for the readers.

2.       The authors have made clear (Line 85) that the mean age reported was for all participants, not just males.